# One-Bath Pretreatment for Enhancing the Color Yield and Anti-Static Properties of Inkjet Printed Polyester Using Disperse Inks

**DOI:** 10.3390/ma12111820

**Published:** 2019-06-05

**Authors:** Hongmei Cao, Li Ai, Zhenming Yang, Yawei Zhu

**Affiliations:** 1College of Textile and Clothing Engineering, Soochow University, Suzhou 215021, China; cao20030305@sina.com (H.C.); aili_aili@126.com (L.A.); yang18896977001@163.com (Z.Y.); 2School of Textile, Changzhou Vocational Institute of Textile and Garment, Changzhou 213164, China

**Keywords:** one-bath pretreatment, inkjet printing, polyester fiber, anti-static property, color yield, copolymerization

## Abstract

This paper presents a simple and economical method for preparing durable anti-static functionalized inkjet prints by using P[St-BA-F6]—novel antistatic agents synthesized by an oxidative polymerization of styrene, butyl acrylate, and allyl alcohol polyether F6. The P[St-BA-F6] was characterized by gel permeation chromatography and Fourier transformation infrared spectroscopy. One bath pretreatment solution containing P[St-BA-F6] and pentaerythritol tetraacrylate (PETA) were applied to polyester fabrics before inkjet printing, in order to enhance the color yield and the anti-static properties. The pretreatment conditions, including the concentrations of P[St-BA-F6], curing temperature, and time, were optimized based on inkjet printed polyester fabrics. SEM (scanning electron microscope), XPS (X-ray photoelectron spectroscopy), XRD (X-ray diffractometer), TG (thermogravimetric), and DSC (differential scanning calorimetry) examined the fabrics. The results showed that the treated PET fabrics exhibited good applied performances, such as higher color yield, better dry rubbing fastness, lower electrostatic voltage, and durable anti-static properties, even after washing 10 times. These results can be attributed to alcohol polythene group (F6) and allyl group (PETA). PETA can be cross-linked with P[St-BA-F6] and PET fiber. The thermal stability of the treated fabric was lower than that of the untreated fabric, owing to the presence of resin film on the fiber surface.

## 1. Introduction

Inkjet printing technology shows many environmental advantages over the conventional dyeing and printing process, which enables the cost-effective short-run for production. The pretreatment process is very important for inkjet printing process. It is because that the diffusion behavior of disperse dye ink and the weave density on untreated polyester (Polyethylene terephthalate (PET)) fabric exerts significant impact on the printing accuracy and quality [1,2].

Plasma treatment is becoming more and more popular for the surface modification of textiles. It only changes the outermost layer of a material without interfering with the bulk properties. Moreover, it offers the advantage of greater chemical flexibility to obtain multifunctional textiles. PET fabric could be modified with atmospheric-pressure air/He plasma or O_2_ plasma surface-treatment to improve the color strength and the pigment adhesion of the treated surfaces [3,4,5,6]. The anti-static behaviors of the plasma treated PET fabrics can also be greatly improved and an acrylic acid treatment can further enhance the anti-static properties of the specimens [7].

In general, the water-soluble xanthan gum is used to modify PET fabrics prior to inkjet printing. It can improve the performance of the aqueous pigment-based inkjet prints and their antibacterial properties [8,9,10], and it can achieve a higher color yield, better sharpness and speed properties, as well as more ecological processes [11]. The UV cured inkjet printed PLA fabrics exhibited good performances such as color fastness and high color strength [12].

An atmospheric pressure plasma treatment or a cross-linking agent were used to impart various new surface characteristics, including hydrophilicity and anti-stativity, to enhance the deposition of printing surface pretreatment polymers in order to improve the final color properties of digital inkjet printer of PET fabric [10,13].

Polyethylene terephthalate (PET) has been widely used in the textile domain, owing to its great mechanical properties, easy processability, and quick drying. However, PET fabric always suffers from comfortability and electrostatic charge due to its poor surface wettability. It was very difficult to improve its hydrophilicity and antistatic properties due to the insufficiency of reactive groups on the PET fabric [14]. Accordingly, it is important to impart PET fabrics with excellent and durable anti-static properties. The surface of a PET fabric could be functionalized with a natural biopolymer [15], an aqueous solution of 3-aminopropyltriethoxysilane (APTES) [16], carboxymethyl chitosan (CMCS) [17], triethylenetetramine (TETA), and tetraethylenepentamine (TEPA) [18]. In addition, it is common to use a binding agent through dip coating to impart durable high electrical conductivity properties, such as SiO_2_, TiO_2_, ZnO, ZrO_2_ nanoparticles, or functionalized organic nanoparticles [19] and using gelatin as a green binding agent [20], and a silver/reduced graphene oxide (Ag/RGO) coating [21] or silver nanoparticles inks for wearable e-textiles [22], as well as a quaternary ammonium salt coating solution [23] and RGO-based wearable e-textiles [24].

Surface modification is the simplest method for improving the conductivity while using poly(2,3-dimethylaniline)/polyaniline composites [25], polymethyl methacrylate (PMMA) substrates [23], but the durability of anti-static properties of modified fabric is not desirable.

This paper aims to explore the possibility for improving the color yield and the anti-static properties of PET fabric by one-bath pretreatment of prior to inkjet printing. Thus, in this study, the P[St-BA-F6] composite was prepared by an oxidative polymerization of butyl acrylate (BA), styrene (St), and allyl alcohol polyether F6 (F6) to expand its potential applications in pretreatment before inkjet printing for antistatic PET fabric. In addition, a cross-linking agent of pentaerythritol tetraacrylate (PETA) was applied in pretreatment solution to improve the durability of P[St-BA-F6] resin film.

## 2. Materials and Methods

### 2.1. Materials

A scoured PET fabric (17.8 tex × 4.4 tex, 73.2 g/m^2^) was purchased in the market. Commercial samples of disperse red ink (D2551 Red) and CYMK inks (D2510 Cyan, D2520 Magenta, D2530 Yellow and P2540 Black) were purchased from DuPont Co., Wilmington, DE, USA. Commercial grade BA, St, and F6 were purchased from Jiangsu Haian Petroleum Chemical Co., Haian, China. Commercial grade emulsifying agent AES (fatty alcohol polyoxyethylene ether sodium sulfate), emulsifying agent TO-7 (heterogeneous 13 alcohol ether), NaOH, and ammonium persulfate were purchased in the market. Commercial grade Sodium carboxymethyl cellulose (CMC) was supplied by Group Chemical Reagent Co., Shanghai, China. Pentaerythritol tetraacrylate (PETA) was purchased from Suzhou Changchunteng Import and Export Co., Suzhou, China.

### 2.2. Synthesis of P[St-BA-F6] Latex

The core-shell particles emulsion containing P[St-BA-F6] were synthesized by seeded emulsion polymerization. They were concretely prepared in two stages, namely core and shell formation. The requisite amounts of emulsifier AES (2.0 g), emulsifier TO-7 (1.0 g), ammonium persulfate (1.0 g), BA (10.0 g), St (3.0 g), and deionized water (33.0 g) were mixed by mechanical stirrer, namely emulsion solutions A (50.0 g). The requisite amounts of emulsion solutions A (33.0 g), F6 (20.0 g) were mixed by a mechanical stirrer, namely emulsion solutions B (53.0 g). The requisite amounts of deionized water (29.0 g) and ammonium persulfate (1.0 g) were mixed in a four-necked round-bottomed flask. The emulsion solutions A (17.0 g) that was placed in a dropping funnel was slowly released into the system at a temperature between 75–80 °C over a period of 30–40 min. Afterwards, emulsion solution B (53.0 g) was slowly dropped into the system at a temperature of 80–85 °C over a period of 90–100 min, with a mixture of shell monomers consisting of BA, St, and F6. Subsequently, the mixture was allowed to react for further 2 h. The core-shell latex particles of P[St-BA-F6] emulsion was obtained (Figure 1). The coagulum contents of P[St-BA-F6] was 94.3%.

### 2.3. Latex Characterization

The latex particles size data of latex particles were obtained from dynamic light scattering on a Malvern particle size (PSD; Nano-ZS90, Malvern Panalytical, Almelo, The Netherlands).

GPC (gel permeation chromatography) analysis was performed in THF (tetrahydrofuran) (1.0 mL/min, 30 °C) while using a Viscotek TDA302 (GPC; Viscotek TDA302, Malvern Panalytical, Almelo, The Netherlands) with a WL. M GPC solvent/sample module.

FTIR (Fourier transform infrared spectroscopy) analyses were obtained from Nicolet 5700 (FTIR; Nicolet 5700, Thermo Electron Co., Waltham, MA, USA) within the frequency range of 650–4000 cm^−1^ by KBr pellets technique and a resolution of 4 cm^−1^ was used to analyze the chemical structure of the dried latex particles.

### 2.4. P[St-BA-F6] Latex Pretreatment and Printing of PET Fabric

Figure 2 shows the preparation process of P[St-BA-F6] latex modified and inkjet printing of PET fabric and crossing reaction on PET fabric. First, a typical recipe prepared the solutions for pretreatment of PET fabric (Table 1). The PET fabric was padded and dried with a different pretreatment recipe by continuous setting and curing machine (M-TENTER, Rabbit Co., Taipei, Taiwan). The fabric was dried at 110 °C for 3 min. Subsequently, PET fabric printing used commercially dispersed dye inkjet ink by the inkjet printer (Stylus Photo R330, Epson, Nagano, Japan). Next, PET fabric was dried at 110 °C for 3 min. and the curing fixation at 170 °C for 90 s for dye fixation and cross-linking reaction. The cured PET fabric was finally washed in alkali solutions (1 g/L NaOH, 1 g/L sodium hyposulfite, 1 g/L anion surfactant LS (3-oleamide-2-methoxy sodium benzenesulfonate), liquor ratio was 1:10) at 70 °C for 15 min. and in diluted at room temperature for 5 min. until all the unfixed dyes and chemicals were removed from the fabric surface.

The following measurements were made in order to evaluate the aimed the printing PET fabric. Color yield measurements of the printed PET fabric were obtained from spectrophotometer (Ultra Scan XE, Hunter-Lab., Reston, VA, USA). The spectrophotometer was set to exclude specular reflection and a large aperture (D_65_ and 10° observer).

The color yield, expressed as a K/S value, a* value, and b* value, with different wavelengths, ranging from 400 to 700 nm within the visible spectrum and measured at 10 nm interval was calculated according to Equation (1).
K/S = (1 − R)^2^/2R(1)where K is equal to absorption coefficient; S is equal to scattering coefficient; and, R is equal to reflectance of the colored sample.

The rubbing fastness test was performed according to the standard (ISO 105-171 X12:2016) using a Model 670 type friction fastness machine (James H. Heal & Co. Ltd., Halifax, UK).

Electrostatic properties measurements of the sample were examined while using the electrostatic attenuation meter (HO110 V2, SSD, Tokyo, Japan). Electrostatic voltage and static half period were obtained from the electrostatic attenuation meter. Fabric samples were conditioned in a standard atmospheric condition (35% ± 5% relative humidity and 20 ± 2 °C) for 24 h prior to the measurement.

The PET fabric was washed 10 times with simulated domestic washing according to the AATCC (American Association of Textile Chemists and Colorists) Test method 135 under the condition of normal washing cycle at 27 ± 3 °C, followed by tumble drying process, in order to determine the durability of the anti-static printed PET fabric.

Scanning electron microscopy (SEM, S-4800, Hitachi, Tokyo, Japan) was used to examine the surface morphology of the PET fabric treated and untreated by P[St-BA-F6]. The magnification of the SEM is 1800 times.

X-ray photoelectron spectroscopy (XPS, ESCALAB 250 XPS, Thermo, Waltham, MA, USA) was used to examine surface chemical composition of fabric surface, using Al Ka radiation (hν = 1486.6 eV) operated at 14.0 kV and 200 W.

Thermogravimetric analysis (TG) was conducted on PET fabric using a thermogravimetric analyzer (TG; G-80, TA Instrument, New Castle, DE, USA). The TG tests were performed in N_2_ atmosphere and a temperature of 50–600 °C at a heating rate of 10 °C/min. and a flow rate of 100 mL/min.

The differential scanning calorimetry (DSC) experiments were conducted in a flowing N_2_ atmosphere on PET fabric. The experiments were conducted in a Perkin-Elmer PE8500 DSC calorimeter (DSC; PE8500, Perkin-Elmer, Waltham, MA, USA). DSC tests were performed in a temperature of −10 to +270 °C at a heating rate of 10 °C/min. and flow rate of 100 mL/min. The sample was heated at 270 °C for 5 min, cooled at a rate of 2 °C/min. to −10 °C during the crystallization process.

The X-ray diffraction (XRD) experiments were conducted on PET fabric using X-ray diffraction (XRD; Empyrean, Malvern Panalytical, Almelo, The Netherland). The samples were recorded with X-ray diffractometer in angle range 0°–60°.

## 3. Results

### 3.1. Structure Characteristic of P[St-BA-F6] Latex Particle

Figure 3 shows the structure characteristic of P[St-BA-F6] core-shell latex particle. Figure 3a presents the size distribution. The mean size of the latex particles is 34 nm. Moreover, a narrow size distribution of the latex particles is demonstrated based on the PDI (particles distribution index) value of 0.347. The size distribution indicates the uniformity of size of the latex’s particles. The uniform size of the latex particle makes it apt for the pretreatment of PET fabric due to adherence of the binder.

Figure 3b presents the results of GPC. The number of average molecular weight Mn and the mass average molecular weight Mw are 1794.5, 4606.9, the molar mass dispersity (Mw/Mn) is 2.57. The Mw/Mn relationship indicative that the molar mass dispersity of the obtained the P[St-BA-F6] core-shell latex particles was at a relatively low level. When analyzing the obtained data, in the main chains of the P[St-BA-F6], it was found that it could be the copolymerization of making 2–3 BA chains, 1 F6 chain, and 2–3 St chains.

Figure 3c presents the results of the FTIR spectrum. It can be seen that the sample shows characteristic peaks at 1643 and 1454 cm^−1^, which are assigned to the C=C stretching of the benzene ring. The peaks at 702 and 619 cm^−1^ are the C-H benzene ring stretching of St [26,27]. The peaks at 3461 and 1731 cm^−1^ are, respectively, assigned to OH stretching and C=O stretching of BA [28]. The peaks at 1116 cm^−1^ are C–O–C asymmetric stretching of allyl alcohol F6 [29]. The peaks at 2869 cm^−1^ are CH_2_ and CH_3_ stretching vibrations, as well as the peak at 1373 and 1249 cm^−1^ are the CH_3_ bending vibrations. In addition, the characteristic peaks at 3040–3010, 1225–1200, and 940–920 cm^−1^, respectively, representing C–C stretching between C=C and C–H stretching, C=C stretching, and O–CH=CH stretching are fairly weak. All of the above-mentioned results indicate that the P[St-BA-F6] composite was successfully fabricated through the oxidative polymerization of BA, St, and F6. 

### 3.2. Effect of P[St-BA-F6]-Treated Fabric on Color Yield and Electrostatic Properties

We use experimental parameter, such as curing fixation temperature, curing fixation time, and P[St-BA-F6] concentration, in order to obtain durable anti-static properties of PET fabric using P[St-BA-F6] pretreatment.

Table 2 and Figure 4 show the electrostatic properties and color yield of the digital inkjet (disperse red ink of D2551-Red) printed fabric treated by P[St-BA-F6] latex pretreatment. The concentration of P[St-BA-F6] was 30 g/L. The curing fixation temperature is 150, 170, 190, and 210 °C, and the corresponding curing fixation time is 120, 90, 45, and 30 s, respectively. In comparison with the untreated PET fabric (Table 2), the electrostatic voltage and static half period are much smaller value when the printed fiber was not washed. When the curing fixation temperature was 150 °C for 120 s, static half period is distinctly increased after washing 10 times from 0.10 s to 26.69 s. It shows the formation of poor durability of P[St-BA-F6] resin film at 150 °C. Furthermore, there has been good anti-static properties after washing, which increases the curing fixation temperature at 170, 190, and 210 °C, respectively. It is better when the curing fixation treatment condition was at 170 °C and 190 °C, where electrostatic voltage and static half period are a litter increase after washing from 0.09–0.13 kV and 0.15–0.24 s to 0.46–0.48 kV and 2.54–4.93 s, respectively.

Figure 4 shows that the curing fixation temperature was 150, 170, 190, and 210 °C, K/S of the prints was 3.58, 10.69, 12.05 and 11.90, respectively. K/S of the prints distinctly increased with increasing the curing fixation temperature at 170–210 °C. When the curing fixation temperature was 170, 190, and 210 °C, the K/S showed an increase by 12.7% and 11.3% when compared with the temperature at 170 °C, respectively. The a* value is almost unchanged, the b* value is a slight increase when the curing fixation temperature was 170–210 °C. These results reveal that the curing fixation temperature is a decisive parameter for the increase of the color yield. K/S was significantly enhanced, especially when the curing fixation temperature and time was 190 °C for 45 s, 210 °C for 30 s, respectively. Therefore, we chose the curing fixation temperature and time at 190 °C for 45 s.

Table 3 and Figure 5 show electrostatic properties and color yield of the digital inkjet (D2551-Red) printed fabric treated or untreated by P[St-BA-F6] latex pretreatment. P[St-BA-F6] concentration was 0, 10, 20, 30, 40, 50 g/L, respectively. The curing fixation was 190 °C for 45 s.

In comparison with untreated PET fabric (Table 3), electrostatic voltage and static half period distinctly decreased with increasing the concentration of P[St-BA-F6]. Electrostatic voltage and static with half period of unwashed fiber increased a little when compared with the 10 times washed fiber. When the P[St-BA-F6] concentration was 30–50 g/L, respectively, the electrostatic voltage and static half period are 0.10–0.13 kV, 0.20–0.24 s for unwashed fiber, 0.42–0.46 kV, 4.54–4.93 s for 10 times washed fiber, respectively. Meanwhile, there has a litter increased anti-static properties of PET fabric after 10 times washing. Therefore, P[St-BA-F6] resin film had excellent durability.

In comparison with the untreated PET fabric (Figure 5), the K/S of the prints increased with increasing P[St-BA-F6] concentration. When the P[St-BA-F6] concentration was 10, 20, 30, 40, and 50 g/L, it can be calculated that the K/S increased by 2.2%, 7.6%, 12.9%, 13.2%, and 13.4%, respectively. K/S was enhanced significantly, especially when the concentration of P[St-BA-F6] was higher than 20 g/L. However, the K/S value is a litter changed with a further increase to 40 and 50 g/L as compared with 30 g/L. The b* and a* value is almost unchanged when the P[St-BA-F6] concentration was 30–50 g/L. These results reveal that the P[St-BA-F6] concentration is a decisive parameter for the increase of K/S value. Therefore, we chose a 30 g/L concentration of P[St-BA-F6].

Table 4 shows that the K/S and color fastness of CMYK (refers to the names of the four ink colors used on the printing press, they are cyan, magenta, yellow and pure black) inks printed on untreated and treated fabric. The concentration of P[St-BA-F6] was 30 g/L. Curing fixation temperature and time was 190 °C for 45 s.

The K/S of treated PET fabric using cyan, magenta, yellow, and black inks was clearly higher than that of the untreated PET fabric, it can be calculated that K/S increased by 25.30%, 33.25%, 40.47%, and 7.47%, respectively. The higher dry rubbing fastness grade was achieved with treated PET fabric using cyan, magenta, and black inks than that with untreated PET fabric. The wet rubbing fastness grade found to be the same for all of the CMYK inks on treated and untreated PET fabric.

Figure 6 shows the wash stability and durability of P[St-BA-F6]-treated PET fabric. The electrostatic voltage is very low during a simulated domestic washing after 0, 5, and 10 washing cycles, respectively (Figure 6a). Figure 6b clearly shows that relatively smooth surfaces are visualized on the untreated fibers. In comparison with the untreated fibers, the Figure 6c SEM photograph demonstrate that a lot of P[St-BA-F6] resin film has been coated on the treated fiber surfaces, and some fiber surface look blunt and block substrate. Moreover, it is distinctly discovered that P[St-BA-F6] resin film has formed an inhomogeneous structure film between the fibers and fibers. According to increase wash times of treated PET fabric by P[St-BA-F6] (Figure 6d,e), it is still distinctly discovered that a litter resin film that is coated on the treated fiber surfaces, although a part of the resin film, fell off after the sample washing, partly because the P[St-BA-F6] resin film on the treated fiber surface is of the molecular dimension and is too thin to be observed by such direct methods. XPS analysis of untreated fabric, P[St-BA-F6]-treated fabric, and printed fabric provide evidence for better wash stability [24]. As seen from wide-scan XPS spectra (Figure 6f), it can be seen almost the same that element C1s (at 283.5 eV) and O1s (at 530.8 eV) were detected on the fiber surface of the three samples. High-resolution C1s XPS analysis that functional groups were same which the element C1s of three samples correspond to characteristics of C–C/C–H (at 284.0 eV), C–O (at 286.0 eV), and O–C=O (at 288.5 eV), respectively (Figure 6g–i) [30,31]. It can be indicated that the cross-linking agent of PETA can be initiated by ammonium persulfate (APS) at proper temperature. Based on above data analysis, it could be summarized that P[St-BA-F6] was predominantly driven by physical interactions, such as Van der Waals force. The interaction between P[St-BA-F6] and PET fabric is weak because P[St-BA-F6] average molecular mass is lower. The formation resin film of P[St-BA-F6] at curing fixation temperature is lacking better durability during post-processing. The cross-linking agent of PETA has excellent chemical reactivity and it can be initiated by APS at proper temperature to the reticulated structure resin film with P[St-BA-F6] and PET fiber. It is obviously enhanced interactions between the resin film and PET fiber. It can see that the anti-static properties to increase are attributed to the alcohol polyether group (hydrophilicity). The durability of anti-static properties to increase is attributed to the allyl group (PETA).

### 3.3. Thermal Behavior on Untreated and Treated Fabrics

Figure 7a shows the thermal decomposition behaviors of the untreated PET fabric and treated PET fabric. When compared to the decomposition temperature, at which 5%, 10% weight loss with untreated fabric and treated fabric, it can be seen that the decomposition temperature of untreated fabric is 399.9 and 411.0 °C, respectively. The decomposition temperature of the treated fabric is 386.5 and 403.3 °C, respectively. The decomposition temperature of treated fabric reduced 13.4 and 7.7 °C, respectively. The main reason for this was the existence of P[St-BA-F6], crossing groups (TEPA), and residual CMC, so the thermal stability of treated fabric decreased. Furthermore, PET fabric was steadier than resin film.

Figure 7b shows DSC behaviors of the untreated PET fabric and treated PET fabric. When DSC scanning form −10 °C to 270 °C, the untreated PET fabrics enthalpy values and melting temperature were 41.08 J/g, 249.2 °C, respectively. The treated PET fabrics did not exhibit any significant difference in enthalpy values (46.11 J/g) and melting temperature (252.5 °C). This indicates that the initial reaction proceeded of crystal melt more difficulty on the treated P[St-BA-F6] PET fabrics, it had higher enthalpy values, onset temperate, and peak temperate. 

When the DSC scanning form 270 °C to −10 °C, the untreated PET fabrics enthalpy values, re-crystallization temperature were −50.65 J/g, 208.3 °C, respectively. The treated PET fabrics also did not exhibit any significant difference in enthalpy values (−50.14 J/g) and re-crystallization temperature (205.6 °C). This indicates that the initial reaction proceeded of re-crystallization more easily on the treated P[St-BA-F6] PET fabrics, it had lower enthalpy values, onset temperate, and peak temperate. 

Figure 7c shows the XRD patterns of the untreated and treated PET fabrics, where no significant difference in three distinctive diffraction peaks at 2θ = 18°, 23°, and 26° in the both XRD curves [32]. This phenomenon indicates that the processing procedures, including preachment and inkjet printing in this study, exert marginal influence on the crystalline structure of PET fabrics. Additionally, this thermal behavior of treated fabric is likely a result of thermal nucleation where some chains or their segments become increasingly parallel as a result of heating [12]. 

## 4. Conclusions

In this study, the nanoscale P[St-BA-F6] core-shell particles that were prepared by way of seeded emulsion polymerization. The addition of P[St-BA-F6] enhanced the color yield and anti-static properties of polyester fabrics. The obtained latex has a mean size of 34 nm and an average of number molecular weight of 1794.5 with narrow size distribution. The FTIR analysis confirmed the successful fabrication of P[St-BA-F6] core-shell particles while using butyl acrylate, styrene, and ally alcohol polyethylene F6. 

The results showed that one-bath pretreatment of polyester fabrics with a P[St-BA-F6] and pentaerythritol tetracycline (PETA) is effective in improving the K/S value and the anti-static properties of the disperse dye inkjet Prints. The optimal pretreatment conditions are as follows: P[St-BA-F6] concentration is 30 g/L, curing fixation temperature is 190 °C, and treatment time was for 45 s.

It was demonstrated that the P[St-BA-F6] treated PET fabrics exhibited higher K/S values, better dry rubbing fastness, and durable anti-static properties, even after 10 times washings. These results can be attributed to the alcohol polythene group (F6) and allyl group (PETA). The SEM measurements indicated that treated polyester fabric was coated with a thin resin film. XPS measurements indicated that PETA can be cross-linked with P[St-BA-F6] and PET fiber. XRD measurements showed the marginal influence on the crystalline structure of PET fabrics. The TG and DSC measurements showed the thermal stability of treated fabric decreased owing to existing resin film. The initial reaction proceeded of crystal melt was more difficult, but the initial reaction proceeded of re-crystallization was more easily compared with the untreated fabrics.

## Figures and Tables

**Figure 1 materials-12-01820-f001:**
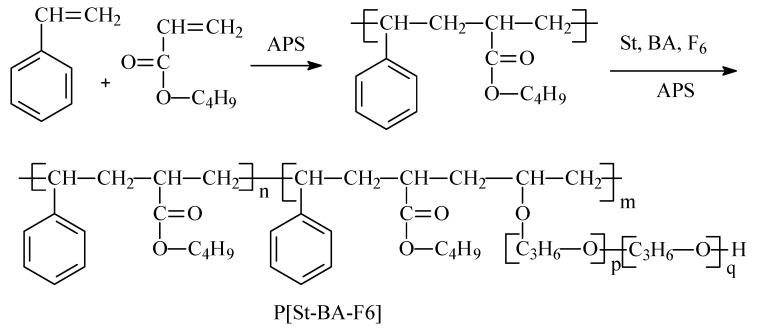
Flowchart of synthesis of P[St-BA-F6].

**Figure 2 materials-12-01820-f002:**
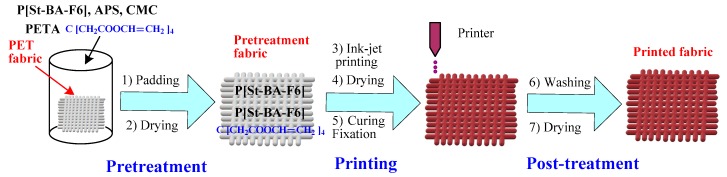
Flowchart of printing process on the pretreated and the crossing reaction of Polyethylene terephthalate (PET) fabric.

**Figure 3 materials-12-01820-f003:**
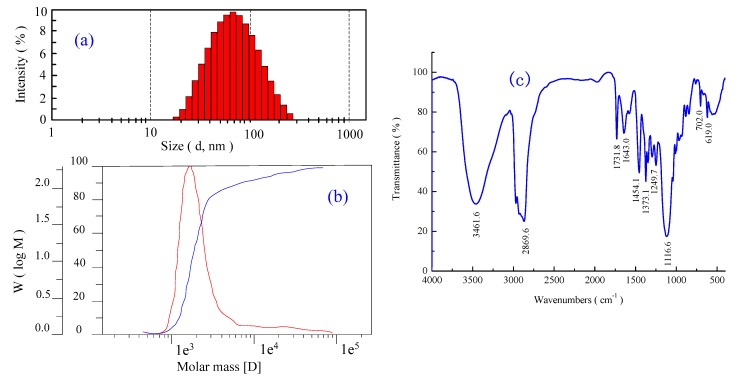
Structure characteristic of P[St-BA-F6] latex particle. (**a**) Size distribution of the P[St-BA-F6] latex particles. (**b**) Gel permeation chromatography (GPC) of P[St-BA-F6] latex. (**c**) Fourier transform infrared spectroscopy (FTIR) spectrum of P[St-BA-F6] composite.

**Figure 4 materials-12-01820-f004:**
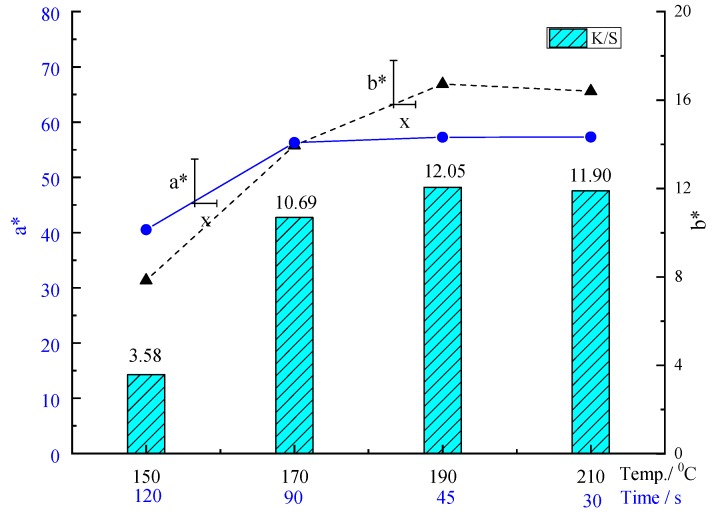
Effect of curing fixation temperature and time on K/S, a* and b* values.

**Figure 5 materials-12-01820-f005:**
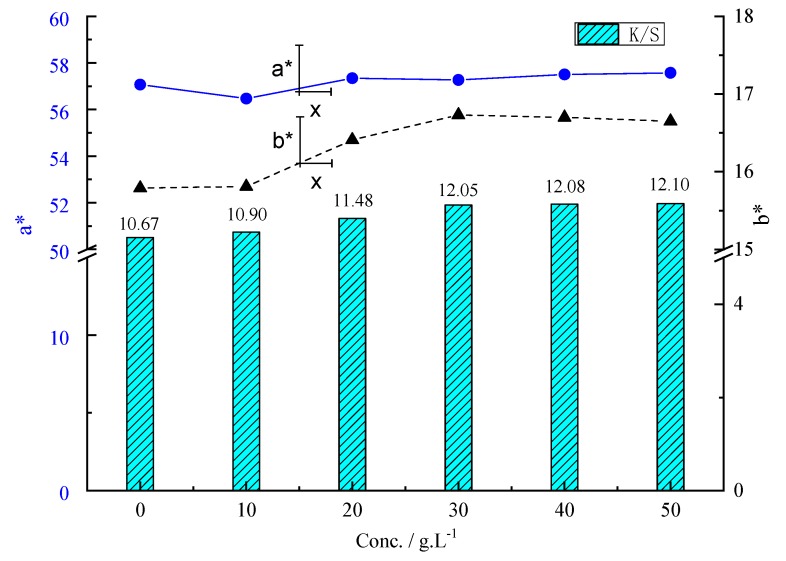
Effect of P[St-BA-F6] concentration on K/S, a* and b* values.

**Figure 6 materials-12-01820-f006:**
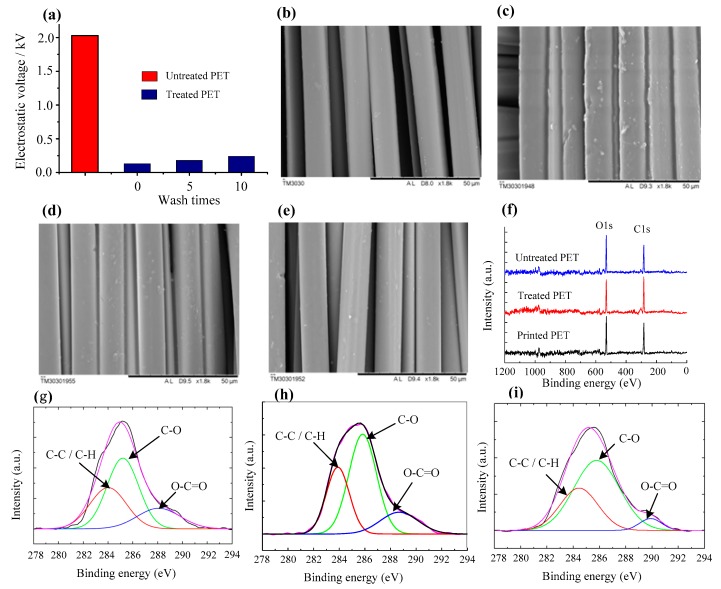
Wash stability and durability of treated PET fabric. (**a**) Change of electrostatic voltage of P[St-BA-F6]-treated fabric with wash times. (**b**) Scanning electron microscopy (SEM) image of untreated (×1800). (**c**) SEM image of P[St-BA-F6]-treated fabric (×1800), wash times = 0. (**d**) SEM image of P[St-BA-F6]-treated fabric (×1800), wash times = 5. (**e**) SEM image of P[St-BA-F6]-treated fabric (×1800), wash times = 10. (**f**) Wide-scan X-ray photoelectron spectroscopy (XPS) spectra of untreated fabric, P[St-BA-F6]-treated fabric, and printed fabric. (**g**) High-resolution C1s XPS spectrum of untreated fabric. (**h**) High-resolution C1s XPS spectrum of P[St-BA-F6]-treated fabric. (**i**) High-resolution C1s XPS spectrum of printed fabric. All the scale bar on SEM images are 50 μm.

**Figure 7 materials-12-01820-f007:**
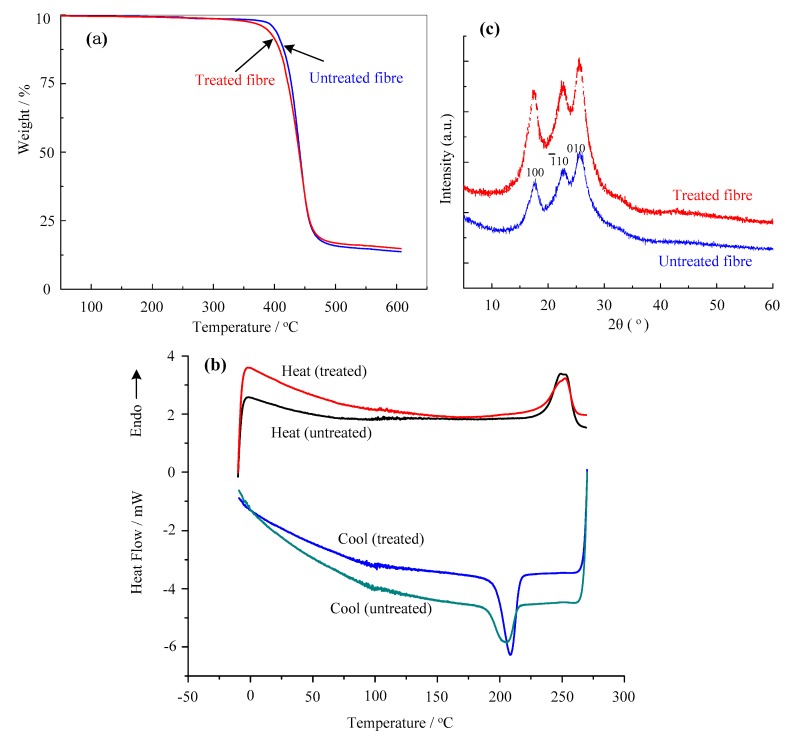
Thermogravimetric analysis (TG), differential scanning calorimetry (DSC), and X-ray diffraction (XRD) of untreated and treated PET fabrics. (**a**) Thermodynamic analysis, (**b**) DSC scanning diagrams, (**c**) XRD patterns.

**Table 1 materials-12-01820-t001:** A typical recipe for solutions for pretreatment of PET fabric.

Functions	Reagent	Weight (g)
Anti-static agent	P[St-BA-F6]	3.0
Thickening agent	CMC	0.6
Penetrating agent	JFC	0.1
Cross-linking	PETA	0.1
Initiator	(NH_4_)_2_S_2_O_8_	0.02
Solvent	H_2_O	96.2

**Table 2 materials-12-01820-t002:** Effect of curing temperature and time on electrostatic voltage and static half period against repeated washings.

Temp./Time	Electrostatic Voltage/kV		Static Half Period/s	
	Wash Times = 0	Wash Times = 10	Wash Times = 0	Wash Times = 10
150 °C/120 s	0.06	0.57	0.10	26.69
170 °C/90 s	0.09	0.48	0.15	2.54
190 °C/45 s	0.13	0.46	0.24	4.93
210 °C/30 s	0.09	0.58	0.56	5.11

**Table 3 materials-12-01820-t003:** Effect of concentration of P[St-BA-F6] on electrostatic voltage and static half period against repeated washings.

Conc./g·L^−1^	Electrostatic Voltage/kV		Static Half Period/s	
	Wash Times = 0	Wash Times = 10	Wash Times = 0	Wash Times = 10
Untreated	2.06	-	>60	-
10	0.32	1.49	0.42	8.21
20	0.20	1.28	0.36	7.38
30	0.13	0.46	0.24	4.93
40	0.12	0.43	0.23	4.54
50	0.10	0.42	0.20	4.79

**Table 4 materials-12-01820-t004:** K/S value and color fastness and properties of inkjet printed on untreated and treated fabric.

Inks	Fabric	K/S	Dry Rubbing	Wet Rubbing
Cyan	untreated	4.27	4	4–5
	treated	5.35	4–5	4–5
Magenta	untreated	4.15	4	4
	treated	5.53	4–5	4
Yellow	untreated	5.56	4–5	4–5
	treated	7.81	4–5	4–5
Black	untreated	10.85	4	4
	treated	11.66	4–5	4

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
