# Peer review of "One-Bath Pretreatment for Enhancing the Color Yield and Anti-Static Properties of Inkjet Printed Polyester Using Disperse Inks"

_materials, 2019, doi:10.3390/ma12111820_

Reviewer 1 Report

The authors present a one-bath pre-treatment of polyester fabrics in order to enhance the colour yield and anti-static properties.  The inkjet-printed fabrics were characterised systematically and the results were discussed. The work is interesting and worth publishing in your journal after addressing following comments:

       Authors mention the benefits of using inkjet printing and an appropriate surface pre-treatment of textiles. Related to this, could authors cite following four papers in their introduction: (a) Karim, M.N., Afroj, S., Rigout, M. et al. J Mater  Sci (2015) 50: 4576. https://doi.org/10.1007/s10853-015-9006-0; b) J. Mater. Chem. C, 2017,5, 11640-11648; c) M.N. Karim et al. / Dyes and Pigments 103 (2014) 168-174 https://doi.org/10.1016/j.dyepig.2013.12.010; and d) arXiv:1905.00839)

2.       I think there are too many texts in materials and methods. it would be good to make it brief and send rest to supporting information.

3.       It would be good to combine Figure 4, 5 and 6 in one figure and explain briefly abou the characterisation of the particle.

4.       Page 7 Line 211 what washing method was used? Could you please add details about the wash powder, time and temperature?

5.       Page 10: XRD analysis doesn’t make any sense to me. Could you please explain the difference between XRD of printed and untreated samples?

6.       Figure 11: Again TGA graphs seem to be same for treated and untreated. Could you please discuss the significance of this result?

7.       Could you please include some XPS data of untreated and surface treated fabrics to demonstrate the effect of one-bath surface pre-treatment? And present XPS data like following papers please: ACS Nano 2019,  13, 4, 3847-3857; DOI: 10.1021/acsnano.9b00319

Overall I would suggest reducing the text of the manuscript and only briefly describe the methodology and results/discussion. Please send additional texts or info to supporting info. Also better presentation of the figures/data would be extremely useful.

Author Response

Response to Reviewer 1 Comments

Manuscript ID: materials-509046

Dear Reviewers,

We first thank you much for reviewing our manuscript. All your comments are fully addressed in the revision. Please refer that the changes made as per your comments are yellow-highlighted in the revised manuscript.

Point 1: Authors mention the benefits of using inkjet printing and an appropriate surface pre-treatment of textiles. Related to this, could authors cite following four papers in their introduction: (a) Karim, M.N., Afroj, S., Rigout, M. et al. J Mater Sci (2015) 50: 4576. https://doi.org/10.1007/s10853-015-9006-0; b) J. Mater. Chem. C, 2017,5, 11640-11648;

c) M.N. Karim et al. / Dyes and Pigments 103 (2014) 168-174https://doi.org/10.1016/j.dye pig.2013.12.010; and d) arXiv:1905.00839)

Response 1: In the revised manuscript, we added references and information. The added statements are:

1) information:

The UV cured inkjet printed PLA fabrics exhibited good performance characteristics such as color fastness and high color strength [12].

In addition, it is common to use a binding agent through dip coating to impart durable high electrical conductivity properties, such as SiO2, TiO2, ZnO, ZrO2 nanoparticles or functionalized organic nanoparticles [19] and using gelatin as a green binding agent [20], and a silver/reduced graphene oxide (Ag/RGO) coating [21] or silver nanoparticles inks for wearable e-textiles [22], as well as a quaternary ammonium salt coating solution [23] and rGO-based wearable e-textiles [24].

2) references:

[12] Karim, MN.; Afroj, S.; Rigout, M.; Yeates, SG.; Carr, C. Towards UV-curable inkjet printing of biodegradable poly (lactic acid) fabrics. Journal of materials science. 2015, 50(13), 4576-4585. doi: 10.1007/s10853-015-9006-0.

[19] Karim, N.; Afroj, S.; Malandraki, A.; Butterworth, S.; Beach, C.; Rigout, M.; Novoselov, KS.; Casson, AJ.; Yeates, SG. All inkjet-printed graphene-based conductive patterns for wearable e-textile applications. Journal of materials chemistry C. 2017, 5(44), 11640-11648. doi:10.1039/c7tc03669h

[22] Karim, N.; Afroj, S.; Tan, S.; Novoselov, KS.; Yeates, SG. All inkjet-printed graphene-silver composite inks for highly conductive wearable E-textiles applications. Applied physics. 2019, arXiv:1905.00839 [physics.app-ph].

[25] Afroj, S.; Karim, N.; Wang, Z.; Tan, S.; He, P.; Holwill, M.; Ghazaryan, D.; Fernando, A.; Novoselov, KS. Engineering graphene flakes for wearable textile sensors via highly scalable and ultrafast yarn dyeing technique. ACS nano. 2019, 13(4), 3847-3857, doi:10.1021/acsnano.9b00319.

[30] Xiao, SL.; Xu, PJ.; Peng, QY.; Chen, JL.; Huang, JK.; Wang, FM.; Noor, N.  Layer-by-layer assembly of polyelectrolyte multilayer onto PET fabric for highly tunable dyeing with water soluble dyestuffs. Polymers. 2017, 9, 735. doi: 10.3390/polym9120735.

[31] Rezaei, F.; Dickey, MD.; Bourham, M.; Hauser, PJ. Surface modification of PET film via a large area atmospheric pressure plasma: An optical analysis of the plasma and surface characterization of the polymer film. Surface & Coatings technology. 2017, 309, 371-371. doi 10.1016/j.surfcoat.2016.11.072.

Point 2:  I think there are too many texts in materials and methods. it would be good to make it brief and send rest to supporting information.

Response 2: In the revised manuscript, we adjust texts in materials and methods. The added statements are:

1) delete texs:

In order to evaluate the aimed core-shell latex particles of P[St-BA-F6] emulsion, the following measurements were made.

The filterable solids obtained from each run were dried. The coagulum contents were calculated using Eq. 1. The coagulum contents of P[St-BA-F6] was 94.3%.

Coagulation% = [W1/W2] × 100%

(1)

where W1 and W2 are the weights of dried filterable solids and initial monomers, respectively.

The silty of latex particles was tested in the following conditions: at room temperature for three months; a diluted solution of 3.0±0.5% P[St-BA-F6] emulsion at room temperature for 72 hours; a mixed solution of P[St-BA-F6] emulsion and 5% NaCl solution (vol./vol. = 1:4) at room temperature for 48 hours; and placed in a centrifugal separator (3000 rpm) for 30 minutes. No breakdown of the latex could be observed in any of these tests.

 A value less than that of the range of PDI equates to a narrower distribution of the particle’s size. The sample was highly diluted (<< span=""> 0.01 wt%) before testing in order to prevent multiple scattering.

The number average molecular weight (Mn) and the mass average molecular weight (Mw) were calibrated with the polystyrene standards.                                         

Figure 3. Flowchart of the crossing reaction between P[St-BA-F6] and PET fabric.

2) combine Figure:

We combine Figure 2 and Figure 3 in one figure (Figure 2).

Point 3: It would be good to combine Figure 4, 5 and 6 in one figure and explain briefly about the characterization of the particle.

Response 3:  In the revised manuscript, we adjust Figure 4, 5 and 6 in one figure (Figure 3).

Point 4: Page 7 Line 211 what washing method was used? Could you please add details about the wash powder, time and temperature?

Response 4: In the revised manuscript, we adjust texts in materials and methods. The added statements are:

In order to determine the durability of the anti-static printed PET fabric, the PET fabric was washed 10 times with simulated domestic washing according to the AATCC Test method 135 under the condition of normal washing cycle at 27 ± 3°C followed by tumble drying process.

Point 5: Page 10: XRD analysis doesn’t make any sense to me. Could you please explain the difference between XRD of printed and untreated samples?

Point 6: Figure 11: Again TGA graphs seem to be same for treated and untreated. Could you please discuss the significance of this result?

Point 7: Could you please include some XPS data of untreated and surface treated fabrics to demonstrate the effect of one-bath surface pre-treatment? And present XPS data like following papers please: ACS Nano 2019,  13, 4, 3847-3857; OI: 10.1021/acsnano.9b00319.

Response 5-7:  In the revised manuscript, we adjust texts in materials and methods. The added statements are:

1) methods:

X-ray photoelectron spectroscopy (XPS, ESCALAB 250 XPS, Thermo, USA) was used to examine surface chemical composition of fabric surface, using Al Ka radiation (hν = 1486.6 eV) operated at 14.0 kV and 200W.

2) combine Figure (refer to ACS Nano 2019, 13, 4, 3847-3857; OI: 10.1021/acsnano. 9b00319)

We adjust Figure 9, XPS Figure and added new Figure (Change of electrostatic voltage of P[St-BA-F6]-treated fabric with wash times) in one figure (Figure 6).

We adjust Figure 10, Figure 11 and Figure 12 in one figure (Figure 7).

    The added statements are:

a) Figure 6 shows the wash stability and durability of P[St-BA-F6]-treated PET fabric. Electrostatic voltage is much low during a simulated domestic washing after 0, 5, 10 times washing cycles, respectively (Figure 6a). Figure 6b clearly shows that relatively smooth surfaces are visualized on the untreated fibers.

b) XPS analysis of untreated fabric, P[St-BA-F6]-treated fabric, and printed fabric provide evidence for better wash stability [24]. As seen from wide-scan XPS spectra (Figure. 6f), it can be seen almost the same as graphite, which that element C1s (at 283.5 eV) and O1s (at 530.8 eV) were detected on the fiber surface of three samples. High-resolution C (1s) XPS analysis that functional groups such as element C1s of three samples which correspond to characteristics of C-C / C-H (at 284.0 eV), C-O (at 286.0 eV), and O-C=O (at 288.5 eV), respectively (Figure. 6g, Figure. 6h and Figure. 6i) [30, 31]. It can be indicated that cross-linking agent of PETA can be initiated by APS at proper temperature.

c) Also, this thermal behavior of treated fabric is likely a result of thermal nucleation where some chains or their segments become increasingly parallel as a result of heating [12].

3) delete texs:

In order to explain briefly about the characterization of TG and DSC, In the revised manuscript, we delete some text, such as onset temperate, the thermal decomposition behaviors at 20%,50% and analysis dates.

Point 8: Overall I would suggest reducing the text of the manuscript and only briefly describe the methodology and results/discussion. Please send additional texts or info to supporting info. Also better presentation of the figures/data would be extremely useful.

Response 8:  According to reviewers’ idea, we have reduced the text of the manuscript. The added statements are:

1) Abstract: The fabrics were examined by SEM (scanning electron microscope), XPS (X-ray photoelectron spectroscopy), XRD (X-ray diffractometer), TG (thermogravimetric) and DSC (differential scanning calorimetry). The results showed that treated PET fabrics exhibited good applied performances such as higher color yield, better dry rubbing fastness, lower electrostatic voltage and durable anti-static properties even after 10 times washings. These results can be attributed to alcohol polythene group (F6) and allyl group (PETA). PETA can be cross-linked with P[St-BA-F6] and PET fiber.

2) conclusions:

a) It was demonstrated that the P[St-BA-F6] treated PET fabrics exhibited higher K/S values, better dry rubbing fastness and durable anti-static properties even after 10 times washings.

b) XPS measurements indicated that PETA can be cross-linked with P[St-BA-F6] and PET fiber. XRD measurements showed the marginal influence on the crystalline structure of PET fabrics.

  We greatly appreciate the Reviewer’s comment. We read through our manuscript again and clean misprints if any.

Reviewer 2 Report

Dear author(s),

Kindly respond the attached comments

Author Response

Dear Reviewers,

We first thank you much for reviewing our manuscript. All your comments are fully addressed in the revision. Please refer that the changes made as per your comments are yellow-highlighted in the revised manuscript.

I suggested its consideration in materials after the mandatory revisions, as following;

Point 1:  I would like to see the crock fastness testing results- important for durability.

Point 2: It could be more preferable if the author can show the color differences between the treated and untreated samples

Point 3: If possible include the results of all the color gamut (CMYK) in addition to the color yield (K/S)-color intensity is the main parameter here.

Response 1-3:  In the revised manuscript, we add the color yield (K/S) and color fastness of CMYK inks printed on untreated and treated fabric. (See Table 4.) The added statements are:

Table 4 shows that the color yield (K/S) and color fastness of CMYK inks printed on untreated and treated fabric. Concentration of P[St-BA-F6] was 30 g/L. Curing fixation temperature and time are 190 for 45 seconds.

The K/S of treated PET fabric using cyan, magenta, yellow and black inks was clearly higher than that of the untreated PET fabric, it can be calculated that K/S increased by 25.30, 33.25, 40.47, 7.47%, respectively. The higher dry rubbing fastness grade was achieved with treated PET fabric using cyan, magenta and black inks than that with untreated PET fabric. The wet rubbing fastness grade found to be same for all the CMYK inks on treated and untreated PET fabric.

Table 4. K/S value and color fastness and properties of inkjet printed on untreated and treated fabric.

Inks

Fabric

K/S

Dry Rubbing

Wet Rubbing

Cyan

untreated

4.27

4

4-5

treated

5.35

4-5

4-5

Magenta

untreated

4.15

4

4

treated

5.53

4-5

4

Yellow

untreated

5.56

4-5

4-5

treated

7.81

4-5

4-5

Black

untreated

10.85

4

4

treated

11.66

4-5

4

We greatly appreciate the Reviewer’s comment. We read through our manuscript again and clean misprints if any.

Reviewer 3 Report

1) In line (202) the O-CH=CH group is mentioned. Where is this group in the chain of P[St-BA-F6]?

2) (264-270) the authors drew a conclusion on the existence of hydrogen and Van der Waals bonds between the fabric and P[St-BA-F6]. The authors came to such conclusion based on what research?

3) Why P[St-BA-F6] was synthesized when the antistatic properties depend only on F6 and PETA?

4) (292-295) - Fig.10 - why is there no signal from the surface layer in the XRD study?

5) What actually results from thermogravimetric studies? Can they be correlated with XRD research in any way?

6) Please, explain the sentence in (337) and (338) lines: `XRD measurements showed a marginal impact on the crystalline structure of PET fabric`

7) (303-305) What do you mean by `silty`?

Author Response

Manuscript ID: materials-509046

Dear Reviewers,

We first thank you much for reviewing our manuscript. All your comments are fully addressed in the revision. Please refer that the changes made as per your comments are yellow-highlighted in the revised manuscript.

Point 1:  In line (202) the O-CH=CH group is mentioned. Where is this group in the chain of P[St-BA-F6]?

Response 1: The O-CH=CH group is existence in PETA. Figure 6 (old) shows film of P[St-BA-F6].

Point 2:  (264-270) the authors drew a conclusion on the existence of hydrogen and Van der Waals bonds between the fabric and P[St-BA-F6]. The authors came to such conclusion based on what research?

Response 2: The P[St-BA-F6] is large molecule polymers. The number of average molecular weight Mn and the mass average molecular weight Mw are 1794.5, 4606.9, respectively. In additional, the benzene ring exists in P[St-BA-F6] and PET fabric. Therefore, it is existence Van der Waals bonds between the fabric and P[St-BA-F6]. But the evidence of hydrogen bond is weak. The adjusted statements are:

it could be summarized that P[St-BA-F6] was predominantly driven by physical interactions such as Van der Waals force.

Point 3:  Why P[St-BA-F6] was synthesized when the antistatic properties depend only on F6 and PETA?

Response 3: The P[St-BA-F6] is large molecule polymers, it contains ester bonds (-O-, hydrophilicity, come from alcohol polyether group (F6)). Therefore, it can be improved the antistatic properties. But the PETA can’t improve the antistatic properties. PETA is a cross-linking agent that increased better wash stability.

Point 4:  (292-295) - Fig.10 - why is there no signal from the surface layer in the XRD study?

Response 4: In order to analysis the surface layer of treated PET, we added XPS experiment. 

Point 5:  What actually results from thermogravimetric studies? Can they be correlated with XRD research in any way?

Response 5: We adjust Figure 10, Figure 11 and Figure 12 in one figure (Figure 7). The added statements are:

Also, this thermal behavior of treated fabric is likely a result of thermal nucleation where some chains or their segments become increasingly parallel as a result of heating [12].

Point 6:  Please, explain the sentence in (337) and (338) lines: `XRD measurements showed a marginal impact on the crystalline structure of PET fabric`

Response 6: In the revised manuscript, we adjust text. The reference is “Tankhiwale, S.; Gupta, MC.; Viswanath, SG. Crystallization studies of crystalline-amorphous blends: Polyethylene terephthalate (PET)-polystyrene (PS). Polymer-plastics technology and engineering. 2002, 41(1), 171-181. Doi:10.1081/PPT-120002068”. The added statements are:

a) This phenomenon indicates that the processing procedures including preachment and inkjet printing in this study exerts marginal influence on the crystalline structure of PET fabrics.

b) XRD measurements showed the marginal influence on the crystalline structure of PET fabrics.

Point 7:  (303-305) What do you mean by `silty`?

Response 7: In the revised manuscript, we adjust texts. The added statements are:

so that the thermal stability of treated fabric decreased. Furthermore, PET fabric was more steady than resin film.

We greatly appreciate the Reviewer’s comment. We read through our manuscript again and clean misprints if any

Round  2

Reviewer 1 Report

The paper has improved significantly after the revision. i think the paper is acceptable; however suggest to add reference 22 from the Scientifc Reports journal  

https://www.nature.com/articles/s41598-019-44420-y

Reviewer 2 Report

Even though all the suggestions did not address (crock fastness), the paper can be accepted as its current form